# Multivariate Multi-Step Long Short-Term Memory Neural Network for Simultaneous Stream-Water Variable Prediction

Marzieh Khosravi [1], Bushra Monowar Duti [2], Munshi Md Shafwat Yazdan [3,*], Shima Ghoochani [4], Neda Nazemi [4] and Hanieh Shabanian [5]

1 Department of Civil and Environmental Engineering, Villanova University, Villanova, PA 19085, USA; mkhosrav@villanova.edu
2 Department of Civil Engineering, East West University, Dhaka 1212, Bangladesh; bushra.duti@ewubd.edu
3 Department of Civil and Environmental Engineering, Idaho State University, Pocatello, ID 83209, USA
4 Department of Civil Engineering, The University of Memphis, Memphis, TN 38111, USA; sghchani@memphis.edu (S.G.)
5 Department of Computer Science, Northern Kentucky University, Highland Heights, KY 41099, USA
* Correspondence: yazdmuns@isu.edu

**Abstract:** Implementing multivariate predictive analysis to ascertain stream-water (SW) parameters including dissolved oxygen, specific conductance, discharge, water level, temperature, pH, and turbidity is crucial in the field of water resource management. This is especially important during a time of rapid climate change, where weather patterns are constantly changing, making it difficult to forecast these SW variables accurately for different water-related problems. Various numerical models based on physics are utilized to forecast the variables associated with surface water (SW). These models rely on numerous hydrologic parameters and require extensive laboratory investigation and calibration to minimize uncertainty. However, with the emergence of data-driven analysis and prediction methods, deep-learning algorithms have demonstrated satisfactory performance in handling sequential data. In this study, a comprehensive Exploratory Data Analysis (EDA) and feature engineering were conducted to prepare the dataset, ensuring optimal performance of the predictive model. A neural network regression model known as Long Short-Term Memory (LSTM) was trained using several years of daily data, enabling the prediction of SW variables up to one week in advance (referred to as lead time) with satisfactory accuracy. The model's performance was evaluated by comparing the predicted data with observed data, analyzing the error distribution, and utilizing error matrices. Improved performance was achieved by increasing the number of epochs and fine-tuning hyperparameters. By applying proper feature engineering and optimization, this model can be adapted to other locations to facilitate univariate predictive analysis and potentially support the real-time prediction of SW variables.

**Keywords:** stream-water; recurrent neural network; Long Short-Term Memory (LSTM); water quality





## 1. Introduction

Surface water has come to be the most crucial resource for societies as a consequence of the growing demand for agriculture, drinking water sources, industrial use, production of electricity, etc. as well as for the necessity to maintain river environmental flow for ecological diversity [1–4]. Increasing climate change has caused harm to both the quality and quantity of surface water, by reduction of freshwater flow due to abrupt changes in rainfall patterns and through increased temperature [5–10]. Stream water (SW) is regarded as the main source of drinking water supply [11]. Often, it is used for recreational purposes such as fishing, swimming, and boating [12–14]. Aquatic life is impacted significantly due to the seasonal changes in water level and discharge over time [15,16]. The demand for water supply, environmental flow, and flood level assessment can all be assessed from discharge and water level through data analysis and numerical modeling, which gives us the scope

of water-quantity estimation [17–19]. Surface water quality on the other hand needs a few more variables to be examined, which are dissolved oxygen, pH, turbidity, toxic substances, and aquatic macroinvertebrate life [20–22]. According to the New Jersey Department of Environmental Protection (NJDEP) (2012) Integrated Water Quality Report, at least one designated use of the aforementioned parameters has below-standard conditions and is termed as Not Supporting (NS) in the Trenton watershed and its sub-catchments [23].

For this study, Dissolved Oxygen (DO), Specific Conductance (SC), temperature, pH, and turbidity are chosen as SW parameters. Temperature, the most important ecological factor, is directly correlated with water's chemical, physical, and biological characteristics [24–27]. It is also the most significant parameter to be affected by climate change [28]. DO is essential for aquatic life to survive, with differing oxygen concentration tolerances among species and life stages. pH and SC have a significant effect on the other metrics of overall water quality, both constructively and adversely. According to previous studies, the positive correlation between them and nitrate ions, ammonia, phosphorus, calcium, and magnesium, or even the detrimental influence of high pH on exotic species invasions, could induce disruptions in natural ecosystems [29–31]. Turbidity is the measure of the relative clarity of water caused by suspended or dissolved particles. High values can significantly reduce the aesthetic quality of streams and influence the natural migrations of species [32]. Assessment of turbidity improves the evaluation and indication of fecal contamination in water bodies such as Escherichia coli, the most common water infection [33,34].

Traditional physics-based numerical models (e.g., HEC-RAS, MIKE) involve spatial and temporal discretization for the entire computational space to compute SW variables, and these require high computational efforts [35,36]. To address the partial differential equation governing the behavior of shallow water in two dimensions, known as the Navier–Stokes equation, various numerical techniques such as Finite Volume, Finite Element, and Finite Difference methods are employed. The Navier–Stokes equation is a comprehensive representation of the conservation of mass, energy, and momentum for incompressible fluids. By utilizing these numerical methods, the equation can be solved, allowing for the analysis and understanding of the dynamics of shallow water systems [37]. The cost of spatial and temporal discretization increases exponentially with the increase in the required resolution and accuracy [38,39]. Input data for the physics-based river models consist of a significant amount of morphological, operational, and measured data. Data preprocessing for physics-based models can be daunting depending on the spatial and temporal tags of the target variables. Physics-based numerical models require measurable and empirical parameters to estimate the target variables [40,41]. Data-informed predictive models provide an efficient alternative approach to forecasting and monitoring both the SW flow and quality assessment of parameters. They offer reduced computational effort while simplifying complicated systems and predict the outcomes using observational data only and excluding complicated physics. Essentially, it is a data-driven model and the characteristics of the data, and their stochastic properties define how the model will behave [42–45]. Lately, Deep Learning (DL), a cutting-edge branch of artificial intelligence, has gained significant popularity as a favored approach for predictive modeling within the realm of water resource management [46]. However, the traditional deep neural network algorithms (e.g., Multilayer Perceptron (MLP)) are limited in their capacity to learn sequential input due to their inability to retain prior knowledge. As a result, it is limited in making accurate predictions for long-term time series, e.g., the temporal distribution of water-table depth is restricted [47,48]. To achieve accurate predictions of the target variables, the Multilayer Perceptron (MLP) algorithm requires intricate data preprocessing procedures [49–51]. Enhancing the MLP model's ability to comprehend the data can be accomplished through data preprocessing techniques, although the involvement of subjective user intervention remains vital. For instance, this includes decisions such as determining the appropriate number of reconstructed components [52]. Numerous reconstructed components need to be calculated, so the preprocessing takes a significant amount of time [53].

Long Short-Term Memory (LSTM) is a unique neural network architecture designed to retain and process extensive sequential data by storing it within a hidden memory cell [54]. LSTM performs well in processing long-term sequential data, utilizing its sophisticated network structure specifically designed to carry the temporal linkage of the time-series data. Water quality and quantity data have not been widely investigated in previous work employing LSTM. The proposed LSTM model only needs a straightforward data preprocessing method, as opposed to the MLP model mentioned earlier [55]. The LSTM neural network exhibits a recurrent nature, with interconnected units forming a directed cycle that enables data to flow in both forward and backward directions within the network. Therefore, the model can preserve past information and use it for future prediction. The LSTM model has been employed extensively in various fields of deep learning, such as speech recognition, natural language processing, automatic image captioning, and machine translation, showcasing its advanced capabilities [47,56,57]. In the realm of water resource forecasting, there have been limited instances where researchers have utilized Recurrent Neural Networks (RNNs) or LSTMs to predict multivariate time-series data [58–60]. The objective of this research is to untangle the pattern of the temporal distribution and the relation among the selected SW (surface water) variables for this study; as well as perform predictive analysis using observed data. To accomplish the goal, a comprehensive Exploratory Data Analysis (EDA) is conducted to investigate the temporal dynamics of the SW variables, and LSTM prediction is performed to predict the future values based on past records. The following sections of the paper demonstrate the study location, data source and collection, EDA, LSTM prediction, performance evaluation, and possible future directions.

This study introduces a novel approach using deep-learning algorithms, specifically the LSTM neural network, to predict stream-water (SW) parameters. By leveraging data-informed analysis and predictive modeling, the research demonstrates the satisfactory performance of LSTM in forecasting SW variables up to one week ahead. Traditional physics-based models are limited by calibration efforts and computational complexity, while the data-informed approach simplifies the prediction process by relying solely on observed data. The LSTM model's ability to process sequential data makes it well-suited for capturing temporal dynamics in SW variables. The study focuses on essential SW parameters such as dissolved oxygen, specific conductance, discharge, water level, temperature, pH, and turbidity, which are crucial for water quality preservation. By successfully applying LSTM models, this research provides an alternative to traditional models, enabling the real-time monitoring and prediction of SW variables. Overall, this study's novelty lies in integrating deep-learning algorithms, comprehensive data analysis, and feature engineering to accurately predict SW variables, opening new possibilities for data-driven approaches in water resource management.

## 2. Data and Methods

### 2.1. Study Area

The monitoring station used in this study is located along the Central Delaware River, in Trenton City, Mercer County, NJ (New Jersey), USA [61]. It is positioned 450 feet upstream of Trenton's Calhoun Street Bridge, 0.5 miles upstream of Assunpink Creek, and 0.9 miles north of Morrisville, PA. The Hydrologic Unit number for this station is 02040105 based on the USGS (United States Geological Survey) water resources database and it is located at 40°13′18″ N, 74°46′41″ W coordinates referenced to the North American Datum of 1983 with the 6780 mi$^2$ of drainage area (Figure 1).

The entire workflow of the EDA and LSTM prediction tasks is divided into three distinct stages. In the first step, data are collected from the USGS web portal, and an exploratory analysis of the SW variables is conducted. To transform the data for training/testing the LSTM algorithm feature engineering is conducted. Variables used in the analysis are listed in Table 1. Activities in the first step are categorized as the transformer. After investigating the dataset and performing data transformation on the variables, the

LSTM neural network is trained using the data prepared in the first step to perform predictive analysis. LSTM neural networks regression model is assessed using several error matrices. These activities are categorized as estimators. The LSTM algorithm is optimized by altering the hyperparameters to reduce the errors in the prediction and achieve satisfactory performance. In the third step, namely the evaluator, the model is deployed to predict the recession rate for a new set of target variables. Model performance is further improved through the iterative incorporation and validation of the input variables.

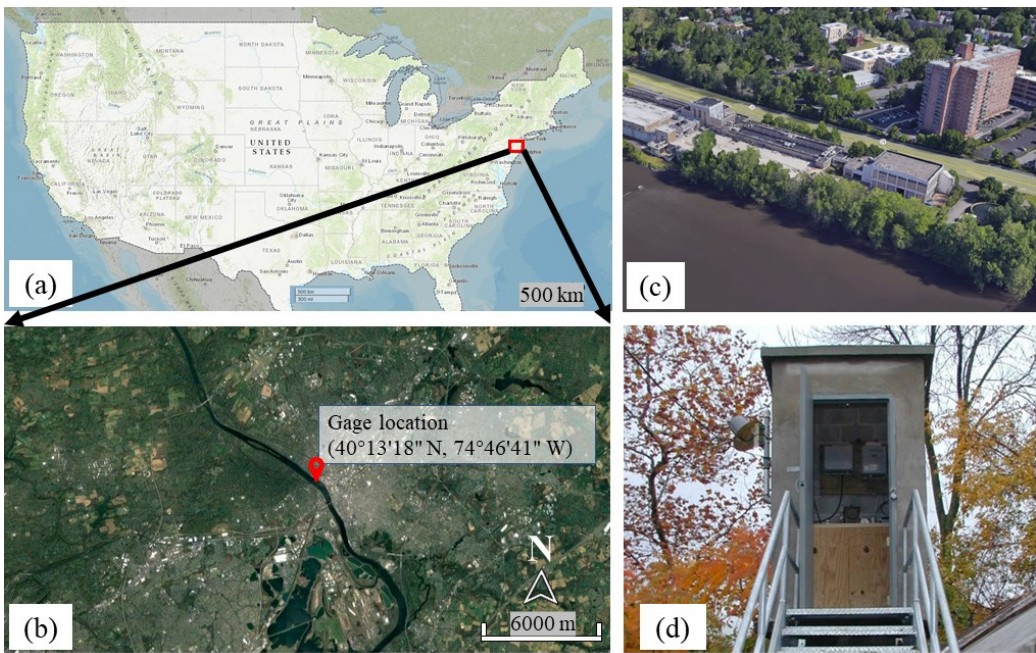

**Figure 1.** Aerial photo of the research location with flow measuring station at Central Delaware (HUC8 02040105). (**a**) the state of the study location is New Jersey state of the United States, (**b**) surrounding topography of USGS gage location of the study, (**c**) surrounding land covers and (**d**) gauge station.

**Table 1.** List of the stream-water variables used for exploratory data analysis and LSTM model.

| SW Parameters | Unit | Descriptions |
|---|---|---|
| Discharge | $ft^3/s$ | Quantity of stream flow |
| Water Level | ft | Stream-water height/level at the gage location |
| Temperature | °C | Sensor-recorded temperature in °C at the gage |
| Dissolved Oxygen (DO) | mg/L | The amount oxygen dissolved in the SW. |
| Turbidity | FNU | Measure of turbidity in Formazin Nephelometric Unit (FNU) |
| pH | - | the acidity or alkalinity of a solution on a logarithmic scale |
| Specific Conductance (SC) | µS/cm | Measure of the collective concentration of dissolved ions in solution |

The LSTM workflow of predicting the SW performance indicator is illustrated in Figure 2. The first step of the workflow is data collection—time-series data of the SW variables are obtained from the USGS National Water Information System: Web Interface [61]. The range of the time-series data for all the variables was different due to the various recorded duration. Mean values of the SW variables are used in this study. The range of the data used in this research is from 25 February 2006 to 8 March 2022 with observed data of approximately 21 years. Historically, from the years 1898 to 1906, peak discharges were measured at Lambertville, NJ, 14.3 miles upstream from the Calhoun Street bridge. The maximum discharge was recorded on 20 August 1955 with the amount of 329,000 $ft^3/s$

and the minimum discharge was 1180 ft$^3$/s, on 31 October 1963. Extreme flooding occurred on 11 October 1903 when the water level reached an elevation of 28.5 ft above the NGVD (National Geodetic Vertical Datum) of 1929, which resulted in a discharge amount of 295,000 ft$^3$/s [61].

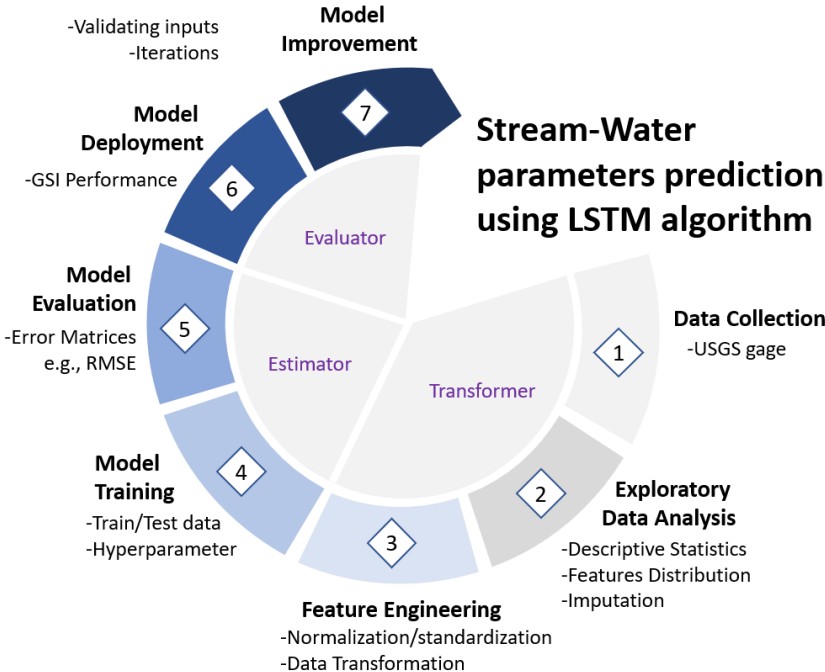

**Figure 2.** Pipeline of the EDA and LSTM prediction tasks illustrates how the activities are linked from the data preprocessing steps to the model deployment stage. The steps are further categorized into their distinct group namely transformer, estimator, and evaluator.

The second step of the workflow is exploratory data analysis, which is covered in Section 2.2. The third step is data transformation (done by feature engineering—discussed in Section 2.3). The fourth step is model training with test data. Finally, the fifth, sixth, and seventh steps of the prediction algorithm are model evaluation, model deployment, and model improvement, respectively.

### 2.2. Multivariate Exploratory Data Analysis (EDA)

In the second activity in Figure 2, a detailed EDA is performed to perceive the attributes and characteristics of the multivariate dataset. To ensure that the LSTM model runs correctly, doing EDA is an essential step in conducting preliminary analyses of the data. Through EDA, a variety of visual techniques and numerical indices are used to study the internal temporal distribution of all the SW variables. To better comprehend the hidden pattern of the distribution of the SW variables, EDA first investigates the variables. This procedure can be further broken down into several steps, including the use of descriptive statistics, the identification of outliers, extreme values, and the verification of normality. Descriptive statistics offer a valuable method for examining the distribution of SW variable values. By considering various statistical measures such as the number of data points, mean, standard deviation, percentiles, interquartile range, and range, one can effectively analyze and characterize the distribution of SW variable values. The complete multivariate descriptive statistics are displayed in Table 2. Histograms with density plots and Pearson's Coefficient of Skewness (PCS) are used to visually display and measure the normality of the values, respectively. The Nearest Neighbors (NN) method was used to find missing data as a numerical imputation method to make the dataset consistent [62].

**Table 2.** Storm-water variables' descriptive statistics.

|  | Count | Mean | Std | Min | 25% | 50% | 75% | Max |
|---|---|---|---|---|---|---|---|---|
| Discharge (ft$^3$/s) | 255,066 | 13,265.43 | 10,657.91 | 2150 | 6240 | 10,800 | 16,100 | 150,000 |
| Water Level (ft) | 255,066 | 9.98 | 1.47 | 7.8 | 8.89 | 9.73 | 10.73 | 20.76 |
| Temperature (°C) | 255,066 | 13.35 | 4.43 | 0 | 12.02 | 13.58 | 15.01 | 31.30 |
| pH | 255,066 | 7.90 | 0.208 | 6.6 | 7.00 | 8.23 | 9.16 | 9.71 |
| SC (µS/cm) | 255,066 | 208.19 | 22.23 | 49 | 201.11 | 208.64 | 221.09 | 453 |
| Turbidity (FNU) | 255,066 | 6.44 | 6.54 | 0.2 | 5.61 | 6.44 | 7.29 | 469 |
| DO (mg/L) | 255,066 | 11.02 | 1.11 | 6 | 11.02 | 11.07 | 12.67 | 16.90 |

Turbidity and other water quality and quantity variables, such as discharge and water level, demonstrate a higher degree of non-normality when compared to other SW variables. A numerical measure of non-normality/skewness, PCS values of discharge, water level, and turbidity are also higher than the PCS values of other SW variables, which indicates relatively less normality.

Figure 3 illustrates the linear connection between two SW variables. Low values of the linear coefficients, delineating the overall non-linearity among several variables, are high. The study has found that the linear correlations can be either positive or negative in direction. A few variables are approximately linearly correlated e.g., discharge and water level, temperature and DO, SC, and pH.

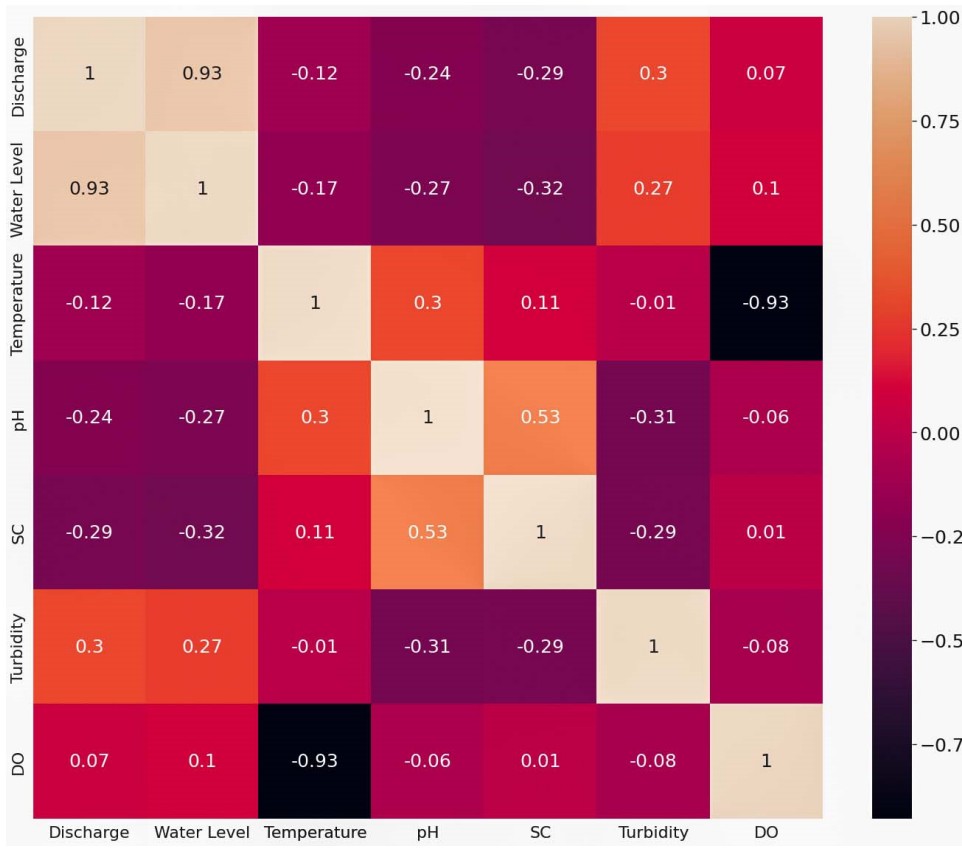

**Figure 3.** Storm-water variables indicated by the correlation heatmaps' bivariate correlation coefficients.

### 2.3. Feature Engineering (FE)

FE is performed after a successful preliminary investigation of the dataset using EDA. The LSTM approach may not produce a satisfactory performance with minimal error without a successful FE. Additionally, without a thorough examination of the inputs, it is impossible to achieve adequate optimization using the iterative gradient descent. Therefore, an extensive feature engineering (FE) process is used to transform the variables into those that most appropriately embody the LSTM learning algorithm [63,64]. FE in this research involves data imputation, data transformation, data standardization, and preparing training, testing, and validation datasets. The process of imputation is used to replace missing data points in a dataset, therefore ensuring its overall consistency. Null values or missing observations were detected in every data series due to issues with the sensor malfunctioning. To address this, the missing values were imputed using the values of the NN (Nearest Neighbor). After a successful imputation, the distribution of the variable series is checked visually and numerically to confirm the normality. PCS (predictability, computability, stability) is used as an indicator of the normality of the variables. Due to the significantly left-skewed distribution and noticeable non-normality of discharge and water depth, neural network regression techniques without the required data transformation do not lead to satisfactory and optimized results [65–67]. Data transformation is performed to decrease the non-normality of discharge and water level. In this research, the performance of LSTMs can be improved using logarithmic transformation to stabilize the variance and lessen the impact of outliers in the feature distribution. In Figure 4, the distributions of the observed and transformed discharge and water level are shown. PCS values increase for all the transformed datasets compared to the original datasets, showing an increase in normality.

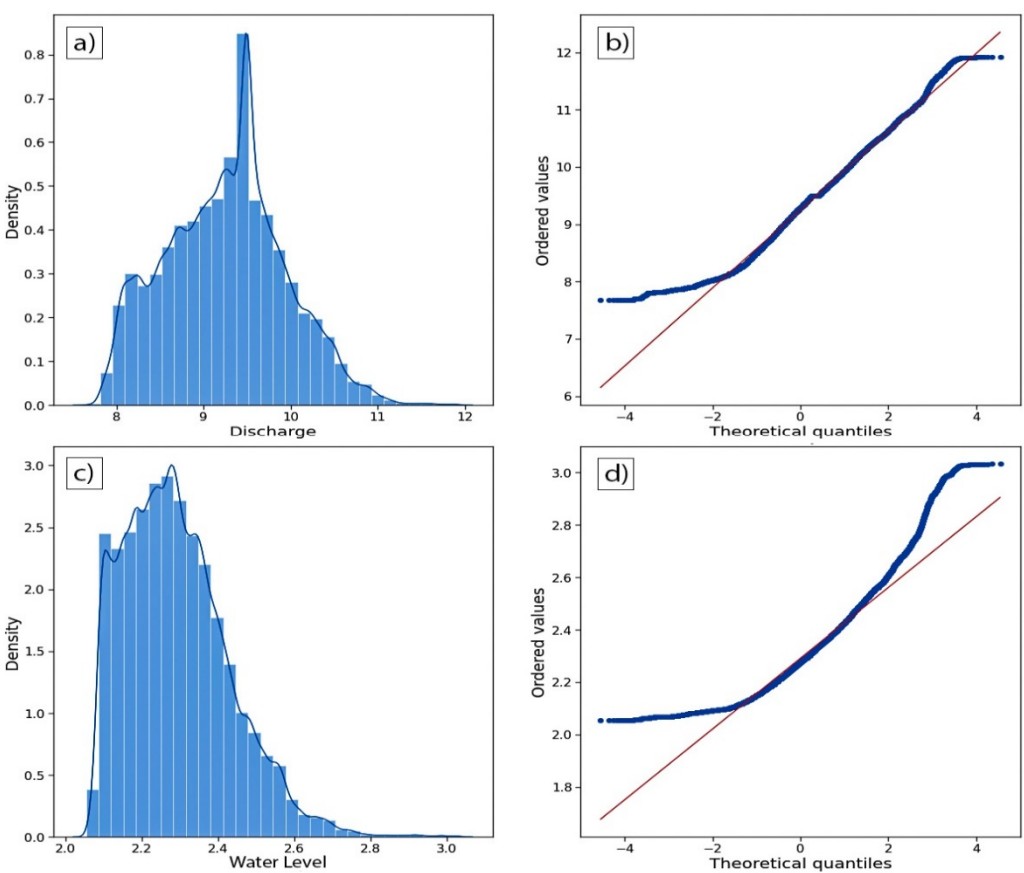

**Figure 4.** Logarithmic transformation is applied to increase the normality of discharge and water level values. (**a**) discharge histogram, (**b**) Q-Q plot of discharge distribution, (**c**) water level histogram and (**d**) Q-Q plot of water level values.

The gradient descent method, in which the step size is influenced by the feature value, is used by the LSTM recurrent neural network. This technique requires updating the steps for all feature values at the same pace to provide a smooth descent towards minima. To establish the LSTM model's training and testing dataset, all the values are normalized utilizing Equation (1).

$$X_{norm} = \frac{X - X_{min}}{X_{max} - X_{min}} \tag{1}$$

*X* denotes the variable of interest and subscript norm, max, and min represent the normalized variable, maximum, and minimum value of the values of the variable. A training set and a testing set were generated from the entire normalized data series to evaluate and test the model.

### 2.4. Long Short-Term Memory (LSTM) Recurrent Neural

LSTM has gained significant popularity as an algorithm for handling time-series data in DL forecasting, specifically when variables rely on past information throughout the series [68,69]. The connections and relationships between variables over a considerable amount of time (long-term) can be captured by the LSTM model [70]. Due to erroneous backpropagation's declining effect, recurrent backpropagation requires a lot of computational power to learn how to maintain long-term data [71]. As a result, RNNs encountered difficulties in accurately capturing long-term relationships, leading to the vanishing gradient problem [72]. LSTM stands out from conventional feedforward neural networks due to the fact it processes and retrieves long-term information because of its feedback connections.

In a typical LSTM algorithm, both long-term memory (c[t − 1]) and short-term memory (h[t − 1]) are processed using numerous gates to filter the data. The memory cell state is updated by forget and update gates for a constant gradient flow [73,74]. Three gates, i.e., input gate $i_g$ (pink), forgot gate $f_g$ (red), and output gate $o_g$ (violet), and the cell state (green) control the flow of information by writing, erasing, keeping track of the past, and reading, respectively. (Figure 5). Therefore, because LSTM can store information over a range of intervals, it is an excellent choice for forecasting time-series data [75]. The forget gate in the LSTM incorporates and filters long-term data, discarding unnecessary information through a processing mechanism. The forget gate filters out redundant data using the sigmoid activation function where the range of the function is 0 and 1 for open and close status, respectively. An LSTM cell's input gate filters and prioritizes incoming data to assess its relevance and importance. The input gate controls the flow of both short-term and long-term information within the cell, similarly to the forget gate, by removing extraneous information using binary activation functions. The value of the upcoming concealed state is also determined by the output gates and is based on the knowledge from preceding inputs.

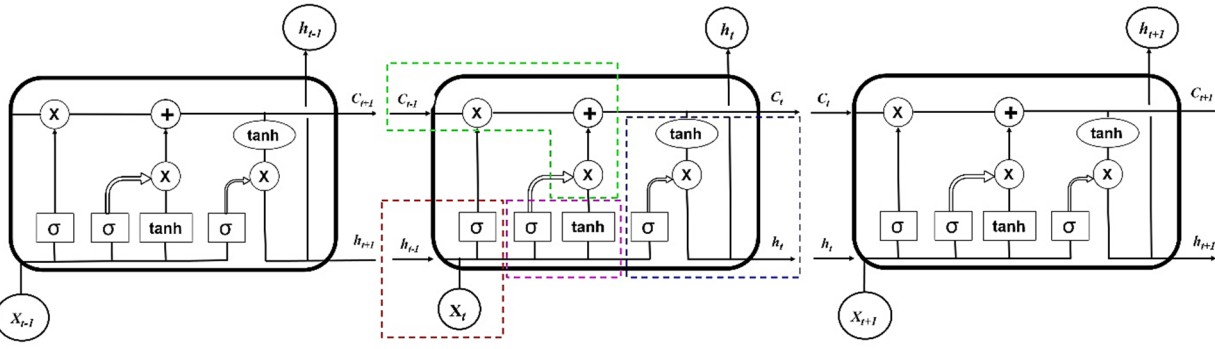

**Figure 5.** Schematic representation of an LSTM architecture.

In this study, a neural network with an LSTM hidden unit accompanied by a dense layer connecting the LSTM target output at the last time-step (t − 1) to a single output neuron with a non-linear activation function. The LSTM model underwent training using the Keras DL library in Python, utilizing the ReLU activation function and employing the coefficient of determination ($R^2$) and Nash–Sutcliffe model efficiency coefficient (E) as loss functions. To predict the SW variable of a time-step in the future, e.g., daily/weekly, values of the variables at previous time steps are used. Hyperparameters are tuned to maximize the performance of the LSTM model through an iterative trial-and-error approach. This work utilized Keras, a Python library that enables the exploration of machine learning algorithms, to determine the optimal set of hyperparameters [76–78]. The impact of several hyperparameters of the LSTM algorithm—specifically, the number of neurons, the batch size, and the size of the epoch—were examined and investigated.

*2.5. Model Evaluation and Improvement*

In the model evaluation step in the fifth activity in Figure 2, the performance of the LSTM model is evaluated using the top three standard error matrices e.g., $R^2$, and the E. Error matrices provide numeric values as the model performance indicator by comparing the observed and predicted values. The $R^2$ value is used to evaluate the LSTM model in showing the model performance improvement. The highest $R^2$ score corresponds to the best predictive accuracy. In addition, the $R^2$ and E are used to illustrate the model response due to the variation in the lead time. The better the model fits the data, the closer the $R^2$ value is to 1. The "E" metric, which is another performance measure in hydrological modeling, is frequently used to assess the model's accuracy [79,80].

A positive E value suggests that the prediction value is a more accurate predictor of the actual recession rate than the average observed value. This is because the anticipated recession rate is more accurate than the average observed value. To ensure that training is effectively fit, it is vital to choose a model's hyperparameters and carefully consider the controlling parameters that impact the learning rate of the model. By optimizing the size of the epoch, batch, and neurons in the stochastic process of the LSTM neural network, the LSTM algorithm's performance in predicting SW parameters is further improved.

## 3. Results and Discussion

LSTM neural networks are employed to forecast the multivariate SW variables. Simultaneous prediction is conducted for various lead-time durations, including 6 h, 12 h, 1 day, 3 days, 1 week, 2 weeks, 3 weeks, and 1 month. The predicted values are then compared to the observed dataset to calculate the error matrices, and $R^2$ and E are used to estimate the error from the predicted SW variables. The incorporation of more epochs results in enhanced model performance. To show the relationship between model efficacy and lead times, error matrices are obtained through numerous models runs. The LSTM model's hyperparameters are modified to effectively optimize model performance, taking into account a set of batch size, epoch size, and the total number of neurons.

*3.1. Predicted and Observed SW Variables*

Figure 6 shows a visual representation of the output from the LSTM algorithm with the observed values of the SW variables. The time-series plots both the observed and predicted values of the SW variables against the number of observations. Because the projected values of SW variables exhibit a distribution that reflects the observed data, the LSTM algorithm functions satisfactorily. The error metrics recorded for all variables and full time-series show LSTM performed well in the case of both training and test sets. In Figure 6, the orange portion of the plot illustrates the training portion of the dataset whereas the green portion shows the testing portion. Dashed lines in blue show the observed data.

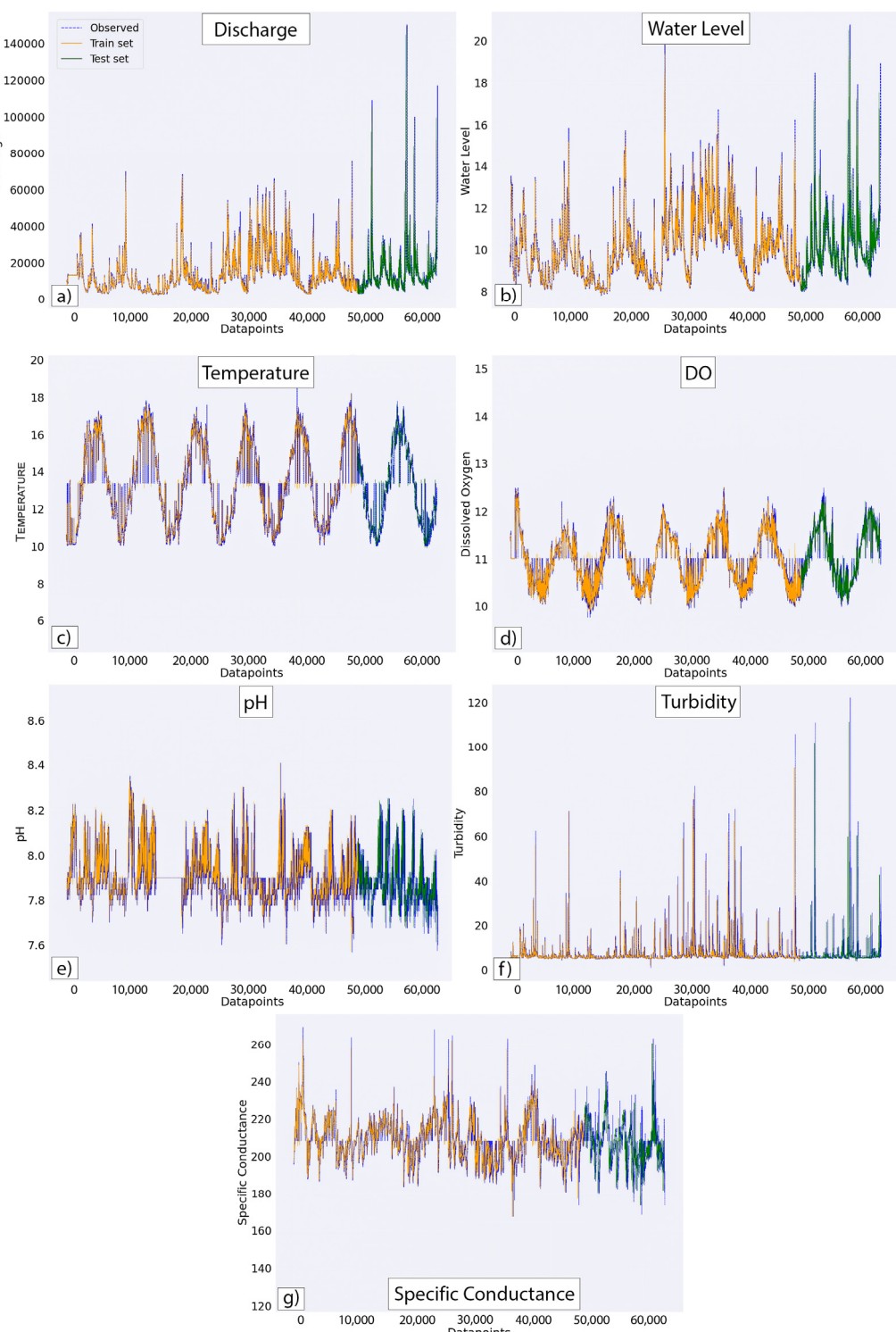

**Figure 6.** Distribution of observed value from the gage records (dashed blue lines) and prediction from the LSTM approach for the stream-water variables, discharge (**a**), water level (**b**), temperature (**c**), DO (**d**), pH (**e**), turbidity (**f**) and SC (**g**) with train/test split (orange).

### 3.2. Model Evaluation Matrices

The LSTM neural network's performance is assessed using three error matrices, namely Root Mean Square Error (RMSE), $R^2$, and E. The model's performance is also evaluated and enhanced by increasing the number of epochs in the neural network. Figure 7 illustrates the change in the error matrix values as the number of epochs increases. The model's performance shows significant improvement from the initial iterations in both the train

and test scenarios. The increase in $R^2$ values reaches a near-steady state after around 15–20 epochs, while there is a slight decrease in performance indicated by an increase in the RMSE value. Additionally, Figure 7 depicts the variation in performance in terms of RMSE with changes in the lead time.

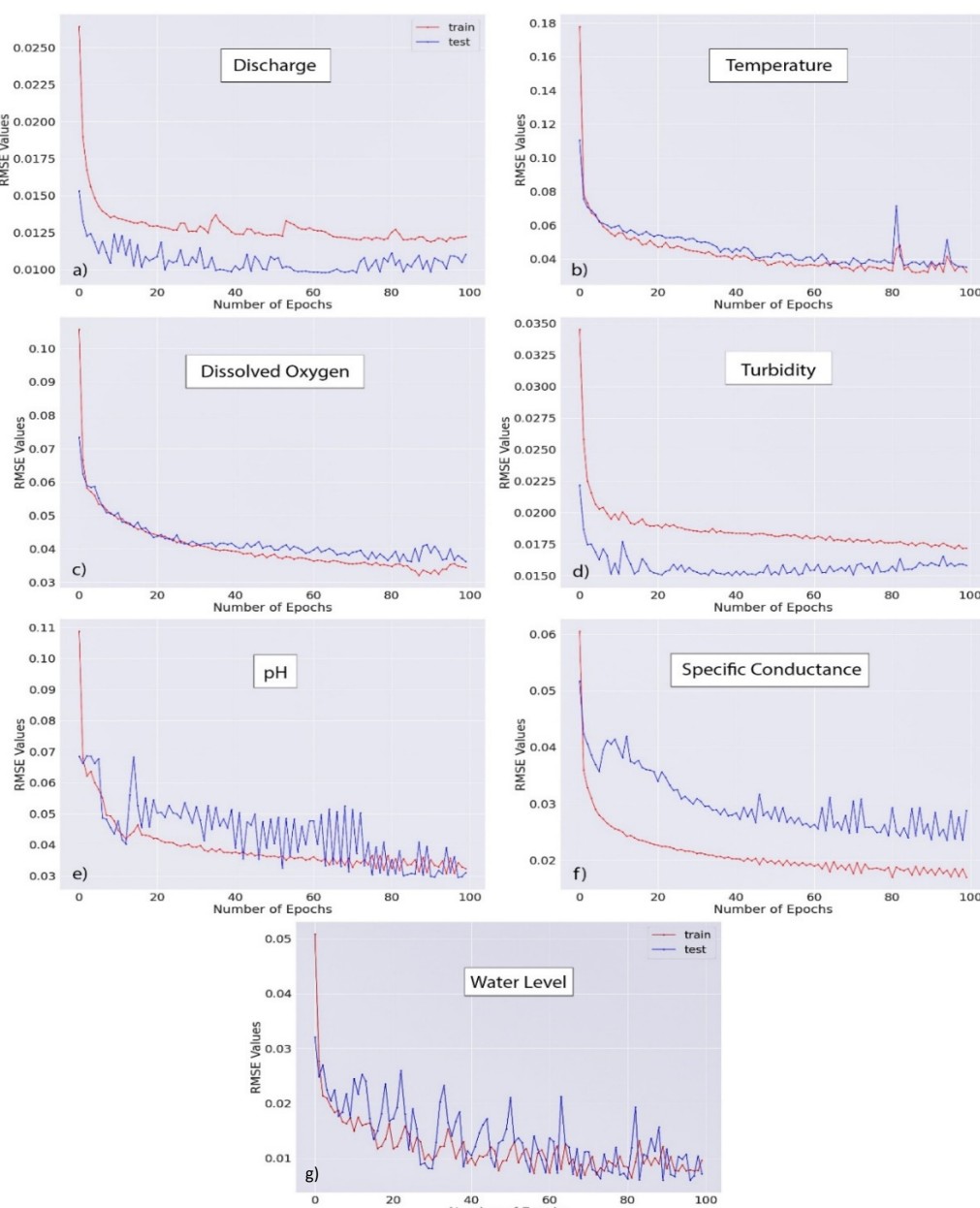

**Figure 7.** Improvement of the model prediction capability with the increase in the number of epochs for the train and test set. RMSE value is the indicator of the model performance for (**a**) discharge, (**b**) temperature, (**c**) dissolved oxygen, (**d**) turbidity, (**e**) pH, (**f**) specific conductance and (**g**) water level.

Error matrices, e.g., $R^2$ and E, are reported for all selected lead times. Lead times are important parameters of the LSTM algorithm toward model performance. Lead-time values are 1 day, 2 days, 3 days, 4 days, 5 days, 6 days, and one week. The values of $R^2$ and E decreases with the increase in lead time, showing the degradation in the model performance with an increase in the lead times (Figure 8). Therefore, the selection of the lead times should be based on model performance and necessity.

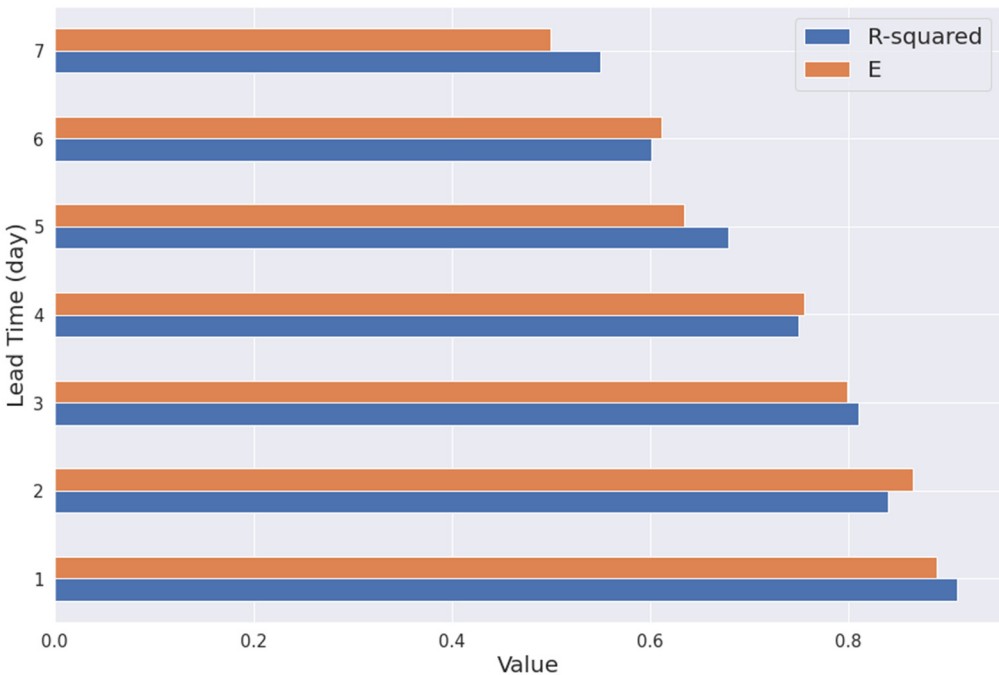

**Figure 8.** Error matrices for various lead times for LSTM neural network model to predict discharge.

Accuracies in the LSTM prediction for all SW parameters are presented in Figure 9 with the help of the coefficient of determination, $R^2$. Observed and predicted values of SW variables from the LSTM prediction are plotted to determine the $R^2$ value. The range of the $R^2$ value for all SW parameters 0.552 to 0.953 delineates satisfactory performance from LSTM model prediction overall. The best prediction with minimum error is found for the DO prediction with the $R^2$ value of 0.953.

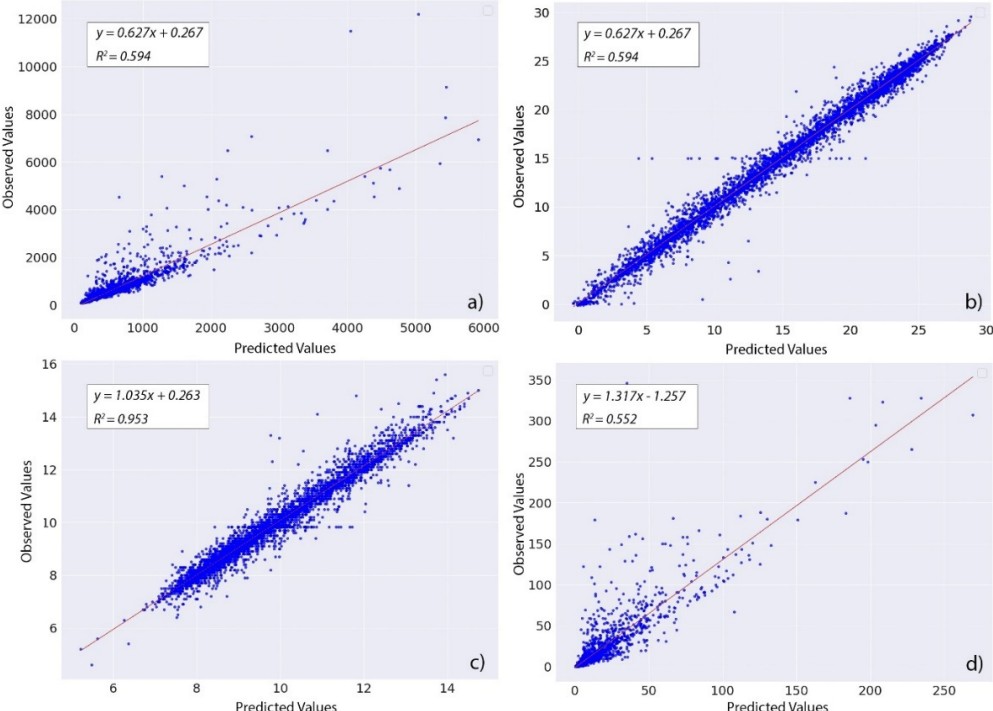

**Figure 9.** *Cont.*

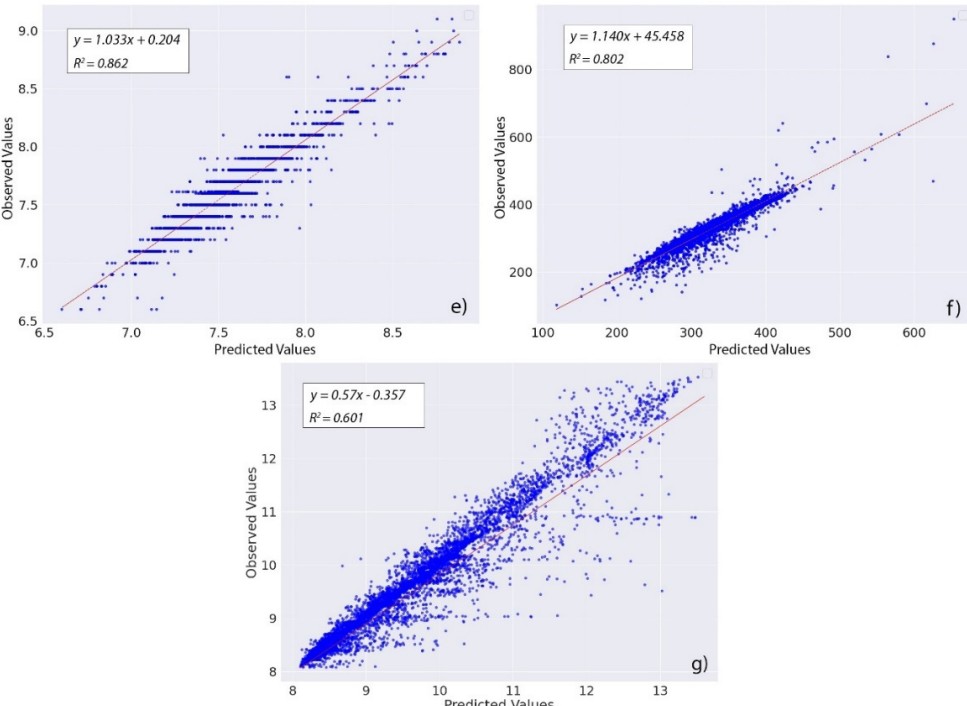

**Figure 9.** Model performances are presented using the scatterplot of the standardized observed and predicted discharge values from the LSTM model and the histogram of the distribution of the difference between the observed and predicted values of the SW parameters. (**a**) Discharge, (**b**) Water Level, (**c**) Temperature, (**d**) Dissolved Oxygen (DO), (**e**) Turbidity, (**f**) pH, (**g**) Specific Conductance (SC).

### 3.3. Hyperparameters Optimization

To ensure optimal model configuration for predictions, it is crucial to perform hyperparameter optimization of the LSTM algorithm. This study focuses on optimizing parameters that specifically impact the SW variables, considering the stochastic nature of the neural network tuning procedure. Initially, the optimization process involves determining the optimal epoch size while keeping the batch size at 4 and using a single neuron. A range of increasing epoch values (50, 75, 100, 125, and 150) is selected to assess the LSTM model's performance. Similarly, a set of batch sizes (1, 3, and 5) and neurons (1 to 5) are chosen with a fixed epoch size of 1000 to observe performance improvements. Further optimization is carried out using an epoch size of 2000, considering the batch size and the number of neurons, which yields the highest $R^2$ value. Figure 10 shows that a batch size of 1 and 5 neurons result in the highest $R^2$ values. Consequently, the optimal combination for the LSTM model prediction is determined to be an epoch size of 2000, a batch size of 3, and a neuron count of 5.

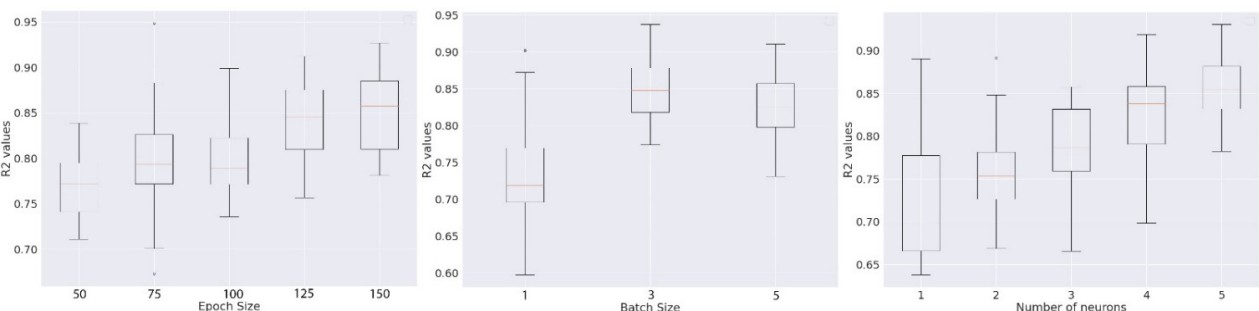

**Figure 10.** Change in the error matrix, $R^2$ values with the increase in the number of epochs batch, and neurons.

## 4. Conclusions

Multivariate prediction of the SW variables under both the water quantity and quality categories at a point location using the observed data can be highly beneficial to water managers and decision-makers to perceive future flooding, irrigation works, and fluvial ecology and aquatic life. Unlike the physics-based numerical models where additional terrain data, land cover/vegetation data, meteorological data, river bathymetry, and human interventions such as structures are pre-requisite as input data, the proposed approach relies only on the previously recorded data of the surface water (SW) variables. The LSTM framework to predict the SW variables can be highly beneficial for the nearby community, where the short-term prediction of the dynamics of the SW variables daily/weekly/monthly in the future plays a critical role. The prediction of water quantity, i.e., discharge and water level, can substantially aid the preparation for flood inundation, irrigation work, and water supply and demand. Prior knowledge of the water quality of the SW can be highly beneficial for aquatic life sustenance, source management of drinking water supply, and irrigation. Water sports activity and water-centric tourism also require a clean aesthetic view of water. The model uses only historical observed data of the variables, and hence it is a data-driven model which utilizes a neural network algorithm to find data patterns and characteristics and predict through stochastic analysis. Several approaches through physics-based numerical modeling techniques have proven inefficient in terms of real-time forecasting and computational efficiency. However, the use of data-informed predictive models is particularly effective at forecasting numerous SW variables without the need for complex differential equations and presumptions. The LSTM algorithm can preserve both the short- and long-term patterns of the time series to forecast.

This study provides a reproducible template for analyzing the distinctiveness of the temporal dynamics of SW variables through comprehensive exploratory data analysis. The distribution of SW variables throughout seven years of data was examined using a variety of modern data exploration tools to find hidden patterns. The proper training of the LSTM algorithm depends on this essential criterion. Utilizing an explicit iterative performance record, the LSTM algorithm was adjusted and refined following a successful training phase. In the same geographic area, SW variables can then be predicted using this improved approach. The algorithm's ability to predict river discharge was shown to be quite effective for the discharge time series, with several error matrices indicating promising performance and little error. The proposed LSTM configuration has been proven to offer satisfactory performance for the SW variables with lead times of up to one week. However, increasing the lead time increases the prediction error, limiting the performance of the LSTM model. Physics-based models are also incompetent in real-time prediction, where the proposed LSTM can easily be coupled with the sensor and cloud to predict the SW variables in real time. Computational time may increase exponentially with the increase in the size of the dataset. Principle parameters obtained after the training process with a minimum error are the numbers of neurons, batch, and epoch size. The parameters optimized to obtain the best LSTM configuration after training the model can be transferable in similar climatic and geographic regions. For instance, if the distribution of the values of the SW variables is identical, e.g., the difference among the PCS values being negligible, the parameters of the trained LSTM model can be transferred and used for predictive analysis in a different location. However, we should not use our LSTM model in an area where the distribution of the feature values through time is dissimilar. The study has a few limitations that should be considered. First, while the LSTM neural network model performed well in predicting surface water variables, it would be beneficial to explore other deep-learning models as well. Models such as CNNs or Transformer-based models may offer alternative approaches and potentially yield different outcomes. Second, the computational efficiency of the LSTM model needs to be considered. Deep-learning models can be computationally demanding, especially with large datasets or complex architectures. Finding ways to optimize model efficiencies, such as using model compression techniques or hardware acceleration, can enhance its practical usability. Additionally, the study focused on a specific range of lead

times, but it would be valuable to assess model performance at longer or shorter lead times. Different lead times present varying prediction challenges and evaluating model reliability across different forecasting horizons is important. Lastly, the generalizability of the LSTM model may be limited to the specific location and dataset used in the study. Adapting and fine-tuning the model for different locations and environmental conditions is necessary to ensure its effectiveness in diverse settings. In summary, while the LSTM model showed promise, exploring other deep-learning models, addressing computational efficiency, considering different lead times, and assessing model generalizability are important for a more comprehensive understanding of the study's findings.

Possible future work includes expanding the study to incorporate ensemble modeling techniques, such as combining multiple deep-learning models, to further improve the accuracy and robustness of the predictions. Additionally, exploring the integration of external environmental factors, such as meteorological data or land use information, could enhance the model's predictive capabilities. Further investigation into the transferability of the LSTM model to different geographical locations and its adaptability to varying hydrological conditions is also crucial. The development of real-time prediction systems using the trained LSTM model, along with the integration of data assimilation techniques, would be valuable for operational applications. Lastly, considering the computational efficiency aspect, exploring model compression techniques or implementing distributed computing frameworks could optimize the model's performance for larger datasets and facilitate its practical implementation in water resource management.

**Author Contributions:** Conceptualization, S.G., M.K. and M.M.S.Y.; methodology, S.G., N.N. and M.K.; software, N.N., M.K., M.M.S.Y. and B.M.D.; validation, N.N.; formal analysis, N.N. and M.K.; investigation, S.G., B.M.D. and N.N.; resources, S.G.; data curation, N.N.; writing—original draft preparation, S.G., N.N. and M.K.; writing—review and editing, S.G., N.N. and M.M.S.Y.; visualization, N.N., M.K. and M.M.S.Y.; supervision, H.S. All authors have read and agreed to the published version of the manuscript.

**Funding:** This research received no external funding.

**Data Availability Statement:** Data collected for the study can be made available upon request from the corresponding author.

**Acknowledgments:** The remarkable assistance provided by Md Abdullah Al Mehedi, a candidate in the Department of Civil and Environmental Engineering at Villanova University, has been instrumental in the completion of this paper and the underlying research. From the initial encounter to the final draft, his enthusiasm, expertise, and meticulous attention to detail have served as a constant source of inspiration, ensuring the progress and quality of this work. Md Abdullah Al Mehedi made significant contributions to various aspects, including model formulation, strategic planning for model simulation, result analysis, and interpretation, as well as identifying future directions and limitations. We express our gratitude for his valuable ideas and contributions. Additionally, we acknowledge the support from the Computer Science program in the School of Computing and Analytics at Northern Kentucky University for facilitating this study.

**Conflicts of Interest:** The authors declare no conflict of interest.

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
