# Peer review of "Multivariate Multi-Step Long Short-Term Memory Neural Network for Simultaneous Stream-Water Variable Prediction"

_2673-4117, doi:10.3390/eng4030109_

Round 1

Reviewer 1 Report

The paper is relatively well-written and addresses an important topic. 

While the model appears to have better performance than traditional physics-based models, its relative performance as compared to similar predictive models should be provided. Also, the advantage of the proposed model in the paper over other similar predictive models are unclear.

ln.158: the range of the data is from 2/25/2006 to 3/8/2022; however, "observed data of seven years" was mentioned. Please clarify it.

Please make sure to spell out acronyms at first mention; e.g., p.3, ln.132: "USGS"; ln.163: "NGVD; Fig. 2: "GSI"; ".  

A careful proofreading and professional editing would significantly improve the quality of the paper. There are many grammar and formatting errors and awkward sentences. E.g., p.1, ln.20: ", ,"; p.1, ln.24: "used using"; p.1, ln.44: "pro-duction"; p.2, ln.49: "purposes e.g.,"; p.2, ln.50-52: "The selected SW quantity variables are discharge and water level of the stream is highly influential on the overbank flooding in the surrounding area, demand of water supply,, and fluvial ecology"; p.2, ln.61: "correlated to"->"correlated with"; p.2, ln.75: "several..." and "equations i.e.,"; p.2, ln.96-98: "it is limited to make accurate predictions for long term time series, such as the temporal distribution of water table depth, is restricted. [47,48]"; p.2, ln.101: "crucial , e.g.,"; p.3, ln.102: "there"; p.3, ln.120: "on the using"; p.4, ln.145: "is   optimized" (please delete the extra spaces); Table 1: "amount (of) oxygen"; ln.182: "the Table"->"Table"; ln.192-3: "Low values of the linear coefficients delineate the overall non-linearity among several variables is high"; ln.195: correlated e.g.,"; ln.210: "observation(s)"; ln.222: "Figure 4(,)"; ln.238: "Neural (Network (RNN))"; ln.278: "(were) examined"; 279: "(I)mprovement"; ln.299: "durations e.g.,"; ln.323: "iterations(;) e.g.,"; ln.328: "performance(;) i.e.," and "Further The"; ln.333: "train(ing)"" and "set(s)"; ln.358: "of   epochs"; ln.359: "(S)imilarly"; ln.363: "  R^2" and "the Figure"->"Figure"; ln:378: quanity(;) i.e.,"; ln.389: "template"->"framework".

Reviewer 2 Report

Some comments:

a. The main objective of the study is well declared.

b. The methodology could be better explained. Considering a profound explanation of the different steps.

c. The novelty of the research for is not described in the current version of the manuscript.

d. In the same way, what about the innovation of the study?

e. Please, include the limitations of the research work.

f. I could not find future research works at the of the conclusion sections, please, include in the revised version of the manuscript.

g. Please, explain the main differences between this manuscript and the followings:

Ghoochani, S., & Nazemi, N. (2023). Simultaneous Prediction of Stream-Water Variables Using Multivariate Multi-Step Long Short-Term Memory Neural Network.

I could find several similarities between both.

h. In my opinion, reconsideration after a detailed explanation of the previous questions. Thank you!

Revise English

Reviewer 3 Report

Thank you for your efforts in this work, and for choosing machine learning technology. However, there are several issues and comments that are needed to be addressed.

1. The papers discuss the usage of a traditional basic LSTM system for prediction, but it misses the novelty if using the algorithm. Please specify what should be the main scientific contribution of this paper.

2. What is the purpose of the Fig.1? and images (c) and (d) specifically to the context of the paper. Additionally, can you please use a proper zooming technical illustration annotation on the images other than using the arrows, it could be very confusing to the reader.

3. in Fig.4, in (b) and (d) what the brown line represents?

4. "The multivariate SW variables are predicted using LSTM neural networks. Multiple 298 lead time durations e.g., 6 hours, 12, hours, 1 day, 3 days, 1 week, 2 weeks, 3 weeks and 1 month are used for simultaneous prediction" why these specific periods were chosen and how often the SW varies per time? is there a specific distribution that the SW follows? what is the Mean and the variance for that?

5. In Figure 6, a visual aid of graphs was used to show the close proximity between the observed and generated points; however, it is clearly seen that almost all the dominant colors of the curve tips are blue, please elaborate.

6. Equation 1 is very primitive and does not add much to the context.

7. The LSTM Model Architecture, and/or the LSTM Model summery is missing and should be discussed in detail.

The English quality is Okay, just several phases, grammar (single and plural) and spacing that need to be taken care of.

Round 2

Reviewer 1 Report

I applaud the authors for addressing my comments.

I applaud the authors for addressing my comments.

Author Response

Thank you. 

Reviewer 2 Report

I have no additional comments

Extensive editing of English language required

Author Response

Thank you.